# An In Vitro Study on the Cytotoxic, Antioxidant, and Antimicrobial Properties of Yamogenin—A Plant Steroidal Saponin and Evaluation of Its Mechanism of Action in Gastric Cancer Cells

**DOI:** 10.3390/ijms25094627

**Published:** 2024-04-24

**Authors:** Justyna Stefanowicz-Hajduk, Piotr Graczyk, Anna Hering, Magdalena Gucwa, Anna Nowak, Rafał Hałasa

**Affiliations:** 1Department of Biology and Pharmaceutical Botany, Medical University of Gdańsk, 80-416 Gdańsk, Poland; petrilasendri21@gumed.edu.pl (P.G.); anna.hering@gumed.edu.pl (A.H.); magdag@gumed.edu.pl (M.G.); 2Department of Cosmetic and Pharmaceutical Chemistry, Pomeranian Medical University in Szczecin, 70-204 Szczecin, Poland; anna.nowak@pum.edu.pl; 3Department of Pharmaceutical Microbiology, Medical University of Gdańsk, 80-416 Gdańsk, Poland; rafal.halasa@gumed.edu.pl

**Keywords:** gastric adenocarcinoma, squamous carcinoma, colorectal carcinoma, neodiosgenin, capecitabine, oxaliplatin

## Abstract

Yamogenin is a steroidal saponin occurring in plant species such as *Asparagus officinalis*, *Dioscorea collettii*, *Trigonella foenum-graecum*, and *Agave* sp. In this study, we evaluated in vitro cytotoxic, antioxidant, and antimicrobial properties of yamogenin. The cytotoxic activity was estimated on human colon cancer HCT116, gastric cancer AGS, squamous carcinoma UM-SCC-6 cells, and human normal fibroblasts with MTT [3-(4,5-dimethylthiazol-2-yl)-2,5-diphenyltetrazolium bromide] assay. The amount of apoptotic and dead AGS cells after treatment with yamogenin was estimated with flow cytometry. Also, in yamogenin-treated AGS cells we investigated the reactive oxygen species (ROS) production, mitochondrial membrane depolarization, activity level of caspase-8 and -9, and gene expression at mRNA level with flow cytometry, luminometry, and RT-PCR, respectively. The antioxidant properties of yamogenin were assessed with DPPH (2,2-diphenyl-1-picrylhydrazyl) and ABTS (2,2′-azino-bis(3-ethylbenzothiazoline-6-sulfonic acid) assays. The antimicrobial potential of the compound was estimated on *Staphylococcus aureus*, *Bacillus cereus*, *Klebsiella pneumoniae*, *Escherichia coli*, *Salmonella enterica*, *Helicobacter pylori*, *Campylobacter coli*, *Campylobacter jejuni*, *Listeria monocytogenes*, *Lactobacillus paracasei*, and *Lactobacillus acidophilus* bacteria strains. Yamogenin showed the strongest cytotoxic effect on AGS cells (IC_50_ 18.50 ± 1.24 µg/mL) among the tested cell lines. This effect was significantly stronger in combinations of yamogenin with oxaliplatin or capecitabine than for the single compounds. Furthermore, yamogenin induced ROS production, depolarized mitochondrial membrane, and increased the activity level of caspase-8 and -9 in AGS cells. RT-PCR analysis revealed that this sapogenin strongly up-regulated *TNFRSF25* expression at the mRNA level. These results indicate that yamogenin induced cell death via the extrinsic and intrinsic way of apoptosis. Antioxidant study showed that yamogenin had moderate in vitro potential (IC_50_ 704.7 ± 5.9 µg/mL in DPPH and 631.09 ± 3.51 µg/mL in ABTS assay) as well as the inhibition of protein denaturation properties (with IC_50_ 1421.92 ± 6.06 µg/mL). Antimicrobial test revealed a weak effect of yamogenin on bacteria strains, the strongest one being against *S. aureus* (with MIC value of 350 µg/mL). In conclusion, yamogenin may be a potential candidate for the treatment and prevention of gastric cancers.

## 1. Introduction

Tumors of the digestive system are classified as those originating from the esophagus, stomach, liver, rectum, and colon. Collectively, they are the most commonly occurring tumors worldwide, accounting for 18.7% of all new cases and 22.6% of cancer-related deaths [1]. Colorectal cancer is the third most common cancer cases in humans. Its occurrence is significantly higher in highly developed countries due to the so-called Western lifestyle—low physical activity, highly processed food, sedentary lifestyle, and the resulting obesity. An increase in cases is also observed in developing countries, such as Eastern European nations. It is estimated that in some highly developed countries, the peak of the cases has already been reached, and the number of the cases will begin to decline (USA and France), while in others, it will remain at the same level (UK and Australia), or even continue to rise (Italy and Spain). Importantly, the increase in the cases does not always translate to an increase in mortality. Intensive screening allows for the detection of the disease at an early stage and provides a chance for a cure [2].

Every twentieth cancer diagnosis in the world is a diagnosis of stomach cancer. Stomach cancer accounts for every thirteenth death due to cancer. The majority of the cases, both in terms of the incidence and mortality, occur in Asian countries. Women statistically suffer from it about half as often as men. For both genders, there is a positive correlation between age and the occurrence. Both environmental and individual risk factors associated with stomach cancer have been identified. Undoubtedly, the most crucial factor is infection with the *Helicobacter pylori* bacterium. It is believed that *H. pylori* is responsible for about 79% of all the cases of stomach cancer and up to 95% of the non-cardiac stomach cancer cases [3]. Other risk factors include, for example, tobacco smoking, which increases the likelihood of developing cancer by 40% for moderate smokers and up to 82% for heavy smokers. Gastroesophageal reflux disease also increases the chances of developing stomach cancer [3].

Despite the variety of gastrointestinal cancer treatment methods, chemotherapy is still the main treatment method. However, the toxic effects of traditional chemotherapy drugs limit their long-term use [4]. New therapeutic agents with high effectiveness and low toxicity are increasingly being developed. Therefore, natural drugs are becoming more and more popular. The plant secondary metabolites are characterized by a variety of biological effects, which may be important not only in the treatment but also in the prevention of many diseases, including cancer [5,6]. One of such substances, which is still little known in terms of anticancer activity, is yamogenin ((25S)-spirost-5-en-3beta-ol, neodiosgenin) belonging to steroid saponins. Numerous studies have confirmed the anticancer effects of steroid saponins [7,8,9]. For example, diosgenin, which has a similar structure to yamogenin, induced apoptosis in the HCT-116 and HT-29 colorectal cancer cell lines [10]. In the azoxymethane (AOM)-induced rodent colon cancer model, diosgenin inhibited the formation of colonic aberrant crypt foci (ACF) and the putative precancerous lesions of the colon [11]. Yamogenin (Figure 1) and diosgenin are stereoisomers and it is possible that their biological activities and mechanism of actions can be comparable. Yamogenin has been identified in a few of the same plants where diosgenin is found, among others, *Trigonella foenum-graecum* [12,13], *Asparagus officinalis* [14,15,16,17]. Previous study showed that yamogenin causes death in ovarian cancer cells, and both the extrinsic and mitochondrial—intrinsic pathways of apoptosis are involved in this process [9].

In this study, we investigated the cytotoxic effect of yamogenin in human cancer squamous UM-SCC-6, colon HCT116, and gastric AGS cells, and estimated the role of the selected cellular factors in the anticancer mechanism of the compound’s action. The antioxidant and antimicrobial activities of this plant metabolite were also evaluated.

## 2. Results

### 2.1. Cytotoxic Effect of Yamogenin and in Combination with Oxaliplatin or Capecitabine

To estimate the cytotoxic effect of yamogenin in HCT116, UM-SCC-6, AGS cells, and fibroblasts, MTT assay was used. The obtained results shown in Figure 2 indicated that the cytotoxic effect of yamogenin was significant in AGS cells, and much weaker in HCT116 cells. The viability of the AGS cells decreased from 100 ± 7.8% for the control to 29.3 ± 3.1% for the yamogenin concentration of 60 µg/mL (IC_50_ value was 18.50 ± 1.24 µg/mL). In the case of HCT116 cells, their viability decreased from 100 ± 7% for the control to 55 ± 3% for the yamogenin concentration of 60 µg/mL. Yamogenin did not exert anticancer effect on UM-SCC-6 cell line. The fibroblasts used as non-cancer control cells were viable at the level of 90% in the concentration range of yamogenin 5–60 µg/mL.

In the next step, we combined yamogenin with oxaliplatin or capecitabine and tested on AGS cells (Figure 3). The results showed that in both cases the combinations of the compounds enhanced the cytotoxic effect on the cells compared to the individual agents (Figure 4). A very significant decrease in the viability of the AGS cells was observed for the combination of oxaliplatin and yamogenin at concentrations above 16 and 40 µg/mL, respectively. The lowest viability of the cells was 20.10 ± 2.84%, 17.33 ± 1.04%, and 5.80 ± 0.33% for the concentrations of oxaliplatin and yamogenin 16 + 40, 24 + 50, and 40 + 60 µg/mL, respectively. The calculated IC_50_ value for this combination was 10.64 ± 0.18 µg/mL. Similarly, a high anticancer effect was observed for the combination of yamogenin with capecitabine at the concentrations of compounds 14.4 + 40, 21.6 + 50, and 36 + 60 µg/mL, respectively. The viability values of the cells were 23.90 ± 1.61%, 20.16 ± 4.18%, and 18.83 ± 2.72%, respectively. The obtained IC_50_ value for this combination was 13.09 ± 1.83 µg/mL.

### 2.2. Yamogenin Increased the Amount of Apoptotic and Dead AGS Cells

The gastric cancer cells were treated with different amounts of yamogenin for 24 h. The data obtained with flow cytometry showed that the compound increased the amount of late apoptotic and dead cells. The significant changes were observed for the highest used concentrations of the compound. The percentage of late apoptotic cells was 21.05 ± 1.24% and 27.69 ± 0.70% for the concentrations of yamogenin 30 and 60 µg/mL, respectively. The amount of dead cells was 18.38 ± 0.40% and 41.77 ± 0.84% for the concentrations of yamogenin 30 and 60 µg/mL, respectively. The percentage of early apoptotic cells did not exceed 1% for all concentration values of yamogenin (Figure 5).

### 2.3. Yamogenin Decreased Mitochondrial Potential in AGS Cells

Flow cytometry was used to estimate the changes in the polarization of the mitochondrial membrane in the AGS cells treated with yamogenin. The results showed that yamogenin caused a significantly decrease in the mitochondrial membrane potential (MMP) at higher used concentrations of yamogenin. The percentage of depolarized live cells was 7.83 ± 1.08%, 19.35 ± 2.31%, and 20.49 ± 2.50% at the compound concentrations of 10, 30, and 60 µg/mL, respectively. The amount of depolarized dead cells was 10.64 ± 1.46, 26.62 ± 2.99, and 51.94 ± 4.05 µg/mL at the compound concentrations of 10, 30, and 60 µg/mL, respectively (Figure 6).

### 2.4. Yamogenin Increased the Level of Reactive Oxygen Species (ROS) in AGS Cells

The level of oxidative stress in gastric cancer cells was estimated with flow cytometry after 24 h of treating the cells with the compound. The results indicate that yamogenin significantly induced oxidative stress at higher used concentrations. The amount of ROS positive (ROS (+)) cells was 14.88 ± 1.27%, 55.89 ± 2.09%, and 75.56 ± 3.36% at the compound concentrations of 10, 30, and 60 µg/mL, respectively (Figure 7).

### 2.5. Yamogenin Induced Cell Cycle Arrest in subG1 Phase in AGS Cells

The AGS cells were treated with yamogenin for 48 h and analyzed with flow cytometry. The obtained results indicate that yamogenin strongly arrested cell cycle in subG1 phase. The percentage of the cells in subG1 phase was 6.80 ± 1.66%, 6.13 ± 0.68%, 9.80 ± 0.42%, and 66.63 ± 1.94% for the control cells (with EtOH 0.75%, *v*/*v*) and the yamogenin concentrations of 10, 30, and 60 µg/mL, respectively. On the other hand, the amount of cells in G0/G1 phase decreased from 28.40 ± 2.10%, 30.15 ± 1.50%, and 30.63 ± 2.17% for the control cells (with EtOH 0.75%, *v*/*v*) and the yamogenin concentrations of 10 and 30 µg/mL, respectively, to 14.5 ± 0.69% for the compound concentration of 60 µg/mL. A similar effect was observed in the case of the cells in S and G2/M phases, where the percentage of cells decreased from 16.85 ± 0.48% and 47.2 ± 1.94% for the control cells (with EtOH 0.75%, *v*/*v*), respectively, to 4.63 ± 0.15% and 13.58 ± 1.62% for the yamogenin concentration of 60 µg/mL (Figure 8).

### 2.6. Yamogenin Triggered Activation of Caspase-8 and -9 in AGS Cells

The gastric cancer cells were treated with yamogenin for 5 and 24 h and the activity level of caspase-8 and -9 was estimated with luminometry. The results showed that yamogenin induced significant changes, especially after 24 h of treating the cells with the compound. The relative activity of caspases-8 and -9 was 1.16 ± 0.14, 1.80 ± 0.06, and 3.64 ± 0.11 and 1.33 ± 0.16, 1.42 ± 0.13, and 3.28 ± 0.06 for the yamogenin concentrations of 30, 60, and 100 µg/mL, respectively. After 5 h, the changes were smaller and the activity level of caspase-8 and -9 was 1.20 ± 0.06, 1.10 ± 0.10, and 1.37 ± 0.02 and 1.40 ± 0.05, 1.31 ± 0.04, and 1.38 ± 0.08 for the compound concentrations of 30, 60, and 100 µg/mL, respectively (Figure 9).

### 2.7. The Effect of Yamogenin on Expression of Genes at mRNA Level in AGS Cells

The gastric cancer cells were incubated with yamogenin for 24 h and gene expression at mRNA level was estimated with RT-PCR. The results showed that yamogenin strongly up-regulated *BCL2A1* (BCL2 related protein A1), *CASP5* (Caspase-5), *DEDD2* (Death effector domain containing 2), *MCL1* (MCL1 apoptosis regulator, BCL2 family member), *NFKBIA* (NF-kappa-B inhibitor alpha), *RELB* (RELB proto-oncogene, NF-KB subunit), and *TNFRSF25* (TNF receptor superfamily member 25). On the other hand, yamogenin significantly down-regulated the mRNA level of 30 genes among 92 (Figure 10). The highest down-regulation we observed was for *BAX* (BCL2-associated X, apoptosis regulator), *BCL3* (BCL3 transcription coactivator), *CASP9* (Caspase-9), *CHUK* (component of the inhibitor of the nuclear factor kappa B kinase complex), *LRDD* (Leucine repeat death domain containing protein), and *TNFRSF1B* (TNF receptor superfamily member 1B).

### 2.8. Antioxidant Activity of Yamogenin and Inhibition of Protein Denaturation

#### 2.8.1. Antiradical Potential of Yamogenin

The antiradical potential of yamogenin was assessed by two tests commonly used to estimate the ability of natural compounds to scavenge free radicals: ABTS (2,2′-azino-bis(3-ethylbenzothiazoline-6-sulfonic acid) and DPPH (2,2-diphenyl-1-picrylhydrazyl). The obtained results are presented as IC_50_ values (µg/mL) in comparison to the standard compound, ascorbic acid. The data summarized in Table 1 indicate the moderate antioxidant activity of yamogenin in both ABTS and DPPH tests (IC_50_ was 704.7 ± 5.9 and 631.09 ± 3.51 µg/mL, respectively).

#### 2.8.2. Inhibition of Protein Denaturation by Yamogenin

The potential to inhibit the inflammation process by yamogenin was assessed with the protein denaturation inhibition assay. In this method, bovine serum albumin (BSA) was used as a protein denaturation model.

The obtained results indicate that yamogenin possessed anti-inflammation dose-dependent properties, reaching IC_50_ 1421.92 ± 6.06 µg/mL. The sapogenin prevented BSA denaturation lesser than diclofenac (IC_50_ 500 µg/mL), which was used as a control.

### 2.9. Antimicrobial Activity of Yamogenin

In this study, we evaluated the antimicrobial activity of yamogenin on different bacteria strains (Table 2). The obtained results indicate that yamogenin had weak antimicrobial effect. The MIC and MBC values were 3.5 mg/mL or higher for almost all bacteria strains, except *S. aureus*, where the MIC value was 0.35 mg/mL.

## 3. Discussion

In this study, we estimated the effect of yamogenin on the human cancer cells of the gastrointestinal tract—gastric AGS, colorectal HCT116, and squamous UM-SCC-6 cell line, isolated from a tumor located at the base of the tongue of a male patient. The tested compound showed diversified cytotoxic activity on the cells; the highest one was observed for the AGS and HCT116 cells and this effect was dose-dependent. We did not observe changes in the viability of the UM-SCC-6 cells as well as human normal fibroblasts, where the results were above 90% at all the used concentrations of yamogenin. Furthermore, the compound was combined with cytostatics—oxaliplatin and capecitabine used in the treatment of gastrointestinal tract cancers. The experiments showed that these combinations enhanced the cytotoxic potential of the compounds in the AGS cells and this effect was dose-dependent. A higher activity on gastric cancer cells was observed in combination with oxaliplatin than with capecitabine. This phenomenon occurred especially at the highest used concentrations of the compounds. In the next step, yamogenin was evaluated towards cell cycle inhibition and the mechanism of action in gastric cancer cells. The mitochondrial potential changes, production of ROS, activation of caspase-8 and -9, and expression of genes at mRNA level involved in cell death were estimated. Yamogenin suppressed cell proliferation and induced a strong inhibition of the cell cycle in the subG1 phase, what indicated that apoptosis was the main way of cell death in gastric cancer cells. To confirm this hypothesis, the experiments showing changes in mitochondrial potential and activity of caspase-8 and -9 were performed. As a result, yamogenin strongly depolarized the mitochondrial membrane and activated caspase-8 and -9.

Apoptosis as one of the main types of regulated cell death is characterized by cellular morphological and biochemical changes. These features are cell shrinkage, the condensation of chromatin, the formation of apoptotic bodies, and DNA fragmentation [19]. These changes were observed after treating the cells with different concentrations of yamogenin. In the apoptosis, two well-known pathways play a key role—receptor/external and internal/mitochondrial [20]. The first one starts from the interaction of cell death receptors (DR) with external factors such as compounds/drugs, radiation, UV, and pathogens. In this pathway, membrane receptors and appropriate ligands are important—FasL/FasR, TNF-α/TNFR1, Apo3L/DR3, Apo2L/DR4, and Apo2L/DR5 [19,21,22]. The first two models—FasL/FasR and TNF-α/TNFR1 are best-described. When these ligands bind to the receptors, the cell death signal is induced and triggers the binding of the FAS-associated death domain (FADD) and TNF-associated death domain (TRADD) protein, respectively [23,24]. Next, the death-inducing signaling complex (DISC) is formed and triggers the activation of procaspase-8 [25]. The external pathway of apoptosis may induce or enhance the intrinsic, mitochondrial way. The protein that connects these two processes is Bid. Its activated form translocates to the mitochondria, interacts with Bcl-2 family proteins, and induces a decrease in the mitochondrial membrane potential (MMP) [20,26,27]. The released apoptotic factors from the space between the inner and outer mitochondrial membrane activate caspase proteases. Mitochondria play an important role in the induction of apoptosis in mammalian cells caused by drugs, DNA damage, oxidative stress, UV radiation, protein kinase inhibition, and growth factor deprivation [28]. In this work, a significant decrease in MMP, the activation of caspase-8 and -9, and an increase in ROS production in AGS cells treated with yamogenin were observed. Additionally, Real-Time PCR analysis revealed that the expression of *TNFRSF* member receptor gene at the mRNA level in the yamogenin-treated AGS cells was significantly up-regulated. All these results indicate that yamogenin triggered cell death by the extrinsic and intrinsic way of apoptosis.

Yao et al. described the impact of the steroid saponin PP9 (pennogenin-3-O-α-L-rhamnopyranosyl-(1→4)-[α-rhamnopyranosyl-(1→2)]-β-D-glucopyranoside) on human colon cancer HCT116 and HT-29 cells. They demonstrated that it arrested the cell cycle in the G2/M phase by increasing the synthesis of the p21 protein and reducing the concentrations of cdc25C, cyclin B1, and cdc2. This was achieved through the inhibition of the PI3K/Akt/GSK3β signaling pathway [29]. Timosaponin AIII, according to Wang et al. induced caspase-dependent apoptosis by the inhibition of XIAP expression (X-linked apoptosis inhibitory protein) in hepatocellular carcinoma HCC cell lines. Moreover, the induction of AMPKα/mTOR signaling led to autophagy and triggered the XIAP lysosomal degradation pathway [30]. In turn, diosgenin—a stereoisomer of yamogenin strongly inhibited the proliferation of laryngocarcinoma HEp-2 and melanoma M4Beu cells, blocked the cell cycle in S and G2/M phases, and activated p53 [31]. The compound induced apoptosis by a mitochondrial pathway in both lines (HEp-2 and M4Beu) with a fall of mitochondrial potential, caspase-9 and -3 activation, the nuclear localization of apoptosis-inducing factor (AIF), and the cleavage of poly (ADP-ribose) polymerase (PARP) [31]. Diosgenin also induced apoptosis in the HCT116 and HT-29 cell lines [10]. In the azoxymethane (AOM)-induced rodent colon cancer model, diosgenin inhibited the formation of colonic aberrant crypt foci (ACF), putative precancerous lesions of the colon, when administered either during initiation/post-initiation or promotion stages. The studies observed that the expression of HMG-CoA reductase at both mRNA and protein levels was significantly lowered by diosgenin. This was accompanied by a decrease in the expression of p21 ras and β-catenin. Furthermore, diosgenin can induce apoptosis in HT-29 cells at least in part by the inhibition of Bcl-2 and induction of caspase-3 [11,32]. Our results obtained in this work are consistent with the previous one, where yamogenin was tested on ovarian cancer cells and triggered both the extrinsic and intrinsic pathway of apoptosis [9].

In the present study, we also tested the antioxidant effect of yamogenin in vitro as well as the ability of the compound to inhibit protein denaturation. According to the literature, the antioxidant capacity of natural compounds can prevent the development of cancer. Similarly, inflammation process is correlated with the cancer formation [33,34,35]. The ability to inhibit protein denaturation by natural products may protect cells against inflammation and hence cancers [36,37]. This research has shown that yamogenin is able to prevent inflammation to some extent; however, further studies are needed in this field. Furthermore, yamogenin showed moderate activity in comparison to ascorbic acid, which indicates that this plant metabolite may be potentially used as an antioxidant agent in the prevention of cancer diseases. So far, yamogenin has not been tested towards anti-inflammatory and antiradical properties. Diosgenin—a stereoisomer of yamogenin was estimated towards scavenging free radicals and exhibited concentration-dependent antioxidant potential. It also increased the enzymatic and non-enzymatic function of the antioxidant network [38]. Moreover, diosgenin derivatives were synthesized and tested in vitro for their antioxidant effect. One of the compounds—p-aminobenzoic derivative revealed 61.6% blocking of the induced lipid oxidation [39].

The antimicrobial test of yamogenin showed weak activity on bacteria strains, including *H. pylori*, which is an important factor in the development of gastric cancers [3]. The strongest potential of the compound we observed was on *S. aureus*. Generally, steroidal saponins have documented activity against different bacteria strains. For example, Spiegel et al. tested dioscin on *H. pylori* and obtained an MIC value of 64 µg/mL [40]. Also, in that study dioscin significantly reduced the formation of *H. pylori* biofilm under Bioflux-generated flow conditions and enhanced the antibacterial activity of commonly used antibiotics (clarithromycin, metronidazole, and levofloxacin). Other in silico study showed that diosgenin and sarsasapogenin were found to be potentially effective in inhibiting the targeted receptors Lpp20 (HP1456) from *H. pylori* and TNF-alpha-inducing protein which reflects their promising role for the treatment of gastric cancer particularly caused by *H. pylori* infection [41]. Diosgenin also showed antimicrobial activity against *S. aureus*, *Pseudomonas aeruginosa*, and *E.coli* (MIC value was 406 µg/mL for *S. aureus* and ≥1024 μg/mL for *P. aeruginosa* and *E.coli*) and potentiating activity in association with gentamicin and ampicillin on *P. aeruginosa* multidrug-resistant bacteria, with norfloxacin against *E. coli* and with gentamicin against *S. aureus* [42]. Fang et al. isolated spirostanol saponins from *Allium tuberosum* and tested on *B. subtilis* and *E. coli.* Tuberosine B showed moderate antibacterial activity on the pathogens [43]. Another steroidal saponin named fruticoside I from *Cordyline fruticosa* leaves was tested on *Enterococcus faecalis* and obtained an MIC value of 128 µg/mL [44]. Steroidal saponins were also isolated from *Paris polyphylla* var. *yunnanensis* and tested on *Propionibacterium acnes*. Chonglouoside SL-6 had significant activity on this bacteria strain [45].

The in vitro study is an initial step before animal and clinical testing. This work is focused only on the in vitro experiments to show at first stage the potential of yamogenin to be used in the treatment of cancer diseases. The obtained data should be complemented by in vivo experiments for the evaluation of the compound’s efficacy and safety in a living organism. In the case of this natural sapogenin, the study will be continued, especially in relation to the anticancer effects and prevention of tumor. Further research will focus on the ability of yamogenin to combat oxidative stress and inflammation as crucial factors of cancer progression.

## 4. Materials and Methods

### 4.1. Preparation of Yamogenin Solution

Yamogenin obtained from Merck Millipore (Burlington, MA, USA) was dissolved in absolute ethanol at the concentration of 8 mg/mL with the use of an ultrasonic water bath (50 Hz for 1 h).

### 4.2. Cell Culture

The human gastric adenocarcinoma AGS and colorectal carcinoma HCT116 cell lines were obtained from the American Type Culture Collection (ATCC, Manassas, VA, USA). The squamous carcinoma UM-SCC-6 cell line and human fibroblasts were obtained from Merck Millipore (Burlington, MA, USA) and LGC Standards (Teddington, Middlesex, UK), respectively. The AGS and UM-SCC-6 cells were cultured in Dulbecco’s Modified Eagle’s Medium DMEM/Ham’s F-12. The HCT116 and fibroblasts were maintained in McCoy’s Medium and Fibroblast Growth Medium with Supplement Mix, respectively. All the media were supplemented with 100 units/mL of penicillin, 100 µg/mL of streptomycin, and 10% (*v*/*v*) fetal bovine serum (FBS) (Merck Millipore, Burlington, MA, USA). The cells were incubated at 37 °C and 5% CO_2_.

### 4.3. MTT Assay

To estimate the cytotoxic effect of yamogenin, MTT [3-(4,5-dimethylthiazol-2-yl)-2,5-diphenyltetrazolium bromide] assay was used [46]. Oxaliplatin and capecitabine were positive controls. All the cell lines were seeded in 96-well plates at a density of 5 × 10^3^ cells/well and treated for 24 h with the plant metabolite at the concentrations of 5–60 µg/mL. The concentration of ethanol did not exceed 0.75% (*v*/*v*). Oxaliplatin and capecitabine were tested in the range of 0.2–40 µg/mL (0.5–100 µM). After treatment, the cells were incubated with MTT (0.5 mg/mL; Merck Millipore, Burlington, MA, USA) for 3 h and then, formazan crystals were dissolved in DMSO. The absorbance of the formazan solution was measured with a plate reader (Epoch, BioTek Instruments, Santa Clara, CA, USA). All the results [±standard deviation (±SD)] were obtained from six repetitions in at least two independent experiments. The data are expressed as IC_50_ values (µg/mL).

### 4.4. Annexin and Dead Cell Assay

To estimate the effect of yamogenin on the viability of the AGS cells, Annexin V and Dead Cell Assay Kit and flow cytometry (Merck Millipore, Burlington, MA, USA) were used [47]. The cells were seeded in 12-well plates (1 × 10^5^ cells/well) and incubated with the compound at the concentrations of 10, 30, and 60 µg/mL. The concentration of ethanol added to the cells did not exceed 0.75% (*v*/*v*). Oxaliplatin was used as a positive control at a concentration of 40 µg/mL. After 24 h, the cells were stained with the kit reagents and analyzed with flow cytometry (Muse Cell Analyzer, Merck Millipore, Burlington, MA, USA). The experiments were performed in three independent repeats.

### 4.5. Cell Cycle Analysis of AGS Cells Treated with Yamogenin

The AGS cells were seeded in a 6-well plate (5 × 10^5^ cells/well) and incubated with yamogenin in the concentration range of 10.0–60.0 µg/mL for 48 h. The concentration of ethanol added to the cells did not exceed 0.75% (*v*/*v*). Oxaliplatin was used as a positive control at a concentration of 40 µg/mL. The cells were prepared with Muse Cell Cycle Assay Kit (Merck Millipore, Burlington, MA, USA) according to the manufacturer’s instruction, and the amount of the cells in each phase of the cell cycle was determined by Muse Cell Analyzer (Merck Millipore, Burlington, MA, USA). The experiment was repeated three times.

### 4.6. Estimation of Mitochondria Depolarization in AGS Cells Treated with Yamogenin

The AGS cells were seeded in a 12-well plate (1 × 10^5^ cells/well) and incubated with yamogenin at the concentrations of 10.0–60.0 µg/mL. The concentration of ethanol added to the cells did not exceed 0.75% (*v*/*v*). Oxaliplatin was used as a positive control at a concentration of 40 µg/mL. After 24 h of the treatment, the cells were stained with Muse MitoPotential Assay Kit (Merck Millipore, Burlington, MA, USA), and the determination of the percentage of depolarized/live and dead cells was conducted with Muse Cell Analyzer according to the manufacturer’s instruction. All the experiments were independently repeated three times.

### 4.7. Reactive Oxygen Species (ROS) Production in AGS Cells Treated with Yamogenin

The AGS cells (1 × 10^5^ cells/well, 12-well plates) were treated with yamogenin in the concentration range of 10.0–60.0 µg/mL. The concentration of ethanol added to the cells did not exceed 0.75% (*v*/*v*). Menadione was used as a positive control at a concentration of 17 µg/mL. After 24 h of incubation, the cells were stained with Muse Oxidative Stress Kit (Merck Millipore, Burlington, MA, USA) and analyzed with Muse Cell Analyzer according to the manufacturer’s instruction. The experiments were performed in three independent repeats.

### 4.8. Caspases-8/9 Activity in AGS Cells Treated with Yamogenin

The caspase-8/9 activity level in the cells was determined with Caspase-Glo 8 or 9 Assay Kit (Promega, Madison, WI, USA) and Glomax Multi+ Detection System (Promega, Madison, WI, USA) according to the manufacturer’s instruction. The cells were seeded in 96-well plates (1 × 10^4^ cells/well), and after 24 h of incubation, they were treated with yamogenin at the concentrations of 30–100 µg/mL for 5 and 24 h. Oxaliplatin was used as a positive control at a concentration of 20 µg/mL. The experiments were performed in three independent repeats.

### 4.9. RT-PCR Analysis of Gene Expression at mRNA Level in AGS Cells Treated with Yamogenin

The AGS cells were incubated with yamogenin at a concentration of 30.0 µg/mL for 24 h. The total RNA of the cells was isolated using the RNeasy Mini Kit (Qiagen, Hilden, Germany) and the concentration of the RNA was estimated with Agilent Technologies 4200 TapeStation (Agilent Technologies, Santa Clara, CA, USA) according to the manufacturer’s protocol. The Maxima First Strand cDNA Synthesis Kit (ThermoFisher Scientific, Waltham, MA, USA) was used for cDNA synthesis.

cDNA was applied on the TaqMan Array Human Apoptosis Fast 96-well plates (ThermoFisher Scientific, Waltham, MA, USA). Each well contained 92 assays for genes associated with cell death and four assays for control genes. The PCR reactions were performed in StepOnePlus Real-Time PCR System (ThermoFisher Scientific, Waltham, MA, USA). The data were obtained in three independently repeated experiments and analyzed with StepOne software v. 2.3 and PermutMatrix software v. 1.9.3 [18].

### 4.10. Antioxidant and Inhibition of Protein Denaturation Assays

#### 4.10.1. Materials

Ascorbic acid, DPPH (2,2-diphenyl-1-picrylhydrazyl), ABTS (2,2′-azino-bis(3-ethylbenzothiazoline-6-sulfonic acid) diammonium salt, potassium persulfate, albumin from bovine serum (BSA), and DMSO (dimethyl sulfoxide) were sourced from the Merck Millipore (Burlington, MA, USA). TRIS-HCl (0.2 M, pH 8) and HPLC-grade methanol were sourced from P.O.Ch. (Gliwice, Poland).

#### 4.10.2. DPPH Assay

The DPPH radical scavenging ability of yamogenin was determined using the spectrophotometric method with ascorbic acid as a positive control [48]. Briefly: 100 μL of different concentrations of yamogenin or ascorbic acid were mixed with 100 μL of 0.06 mM DPPH methanolic solution and incubated at room temperature in the dark for 30 min. The change in absorbance at λ = 517 nm was analyzed with a 96-well microplate reader (Epoch, BioTek Instruments, Santa Clara, CA, USA). The control was composed of DPPH and absolute ethanol.

DPPH inhibition was calculated according to the following equation:DPPH inhibition (%) = [(A_control_ − A_sample_)/A_control_] × 100%

The radical scavenging activity of the samples was shown as the IC_50_ value (the concentration of the analyzed compound that caused a decrease in the non-reduced form of the DPPH radical by 50%). The experiment was performed in three independent analysis, three repetitions in each (*n* = 9).

#### 4.10.3. ABTS Assay

The ABTS radical scavenging assay of yamogenin was conducted using the spectrophotometric method with ascorbic acid as a positive control [48]. Briefly: 30 μL of different concentrations of yamogenin or ascorbic acid were mixed with 170 μL of ABTS solution (2 mM ABTS diammonium salt, 3.5 mM potassium persulfate) and completed with water to a final volume of 300 μL. After 30 min of incubation at 30 °C in the dark, the change in absorbance was observed at λ = 750 nm by a 96-well microplate reader (Epoch, BioTek Instruments, Santa Clara, CA, USA). The control was composed of ABTS solution and absolute ethanol.

ABTS inhibition was calculated according to the following equation:ABTS inhibition (%) = [(A_control_ − A_sample_)/A_control_] × 100%

The radical scavenging activity of the samples was shown as the IC_50_ value (the concentration of the analyzed compound that caused a decrease in the non-reduced form of the ABTS radical by 50%). The experiment was performed in three independent analysis, three repetitions in each (*n* = 9).

#### 4.10.4. Inhibition of Protein Denaturation

The assessment of protein denaturation inhibition was conducted using the spectrophotometric method [37]. Diclofenac (500 µg/mL) was used as a positive control. The assay is based on assessment of the possibility of inhibiting the denaturation of bovine serum albumin (BSA) by yamogenin. Briefly: 50 µL of yamogenin dilutions were mixed with 45 µL of 5% aqueous BSA solution (*v*/*v*) and 140 µL of phosphate-buffered saline (PBS, pH 6.4). The mixtures were incubated at 37 °C for 15 min; afterwards, the samples were heated at 70 °C for 5 min and then cooled on ice to 25 °C. The change in absorbance was observed at λ = 660 nm in a 96-well microplate reader (Epoch, BioTek Instruments, Santa Clara, CA, USA). Distilled water was used as a control sample.

The inhibition of protein denaturation was calculated using the following equation:Denaturation inhibition (%) = [(1 − A_sample_)/A_control_] × 100%

The experiment was performed in three independent analysis, three repetitions in each (*n* = 9).

### 4.11. Antimicrobial Activity of Yamogenin

Microorganisms: *Staphylococcus aureus* ATCC6538, *Bacillus cereus* ATCC11778, *Klebsiella pneumoniae* ATCC13883, *Escherichia coli* ATCC8739, *Salmonella enterica* ATCC13076, *Helicobacter pylori* ATCC43504, *Campylobacter coli* ZMF (collection of the Department of Pharmaceutical Microbiology, Medical University of Gdańsk, Gdańsk, Poland), *Campylobacter jejuni* ZMF (collection of the Department of Pharmaceutical Microbiology, Medical University of Gdańsk, Gdańsk, Poland), *Listeria monocytogenes* PCM2191, *Lactobacillus paracasei* PCM2639, and *Lactobacillus acidophilus* PCM2499 were used in this study.

*Listeria monocytogenes* PCM2191 strain grew in Brain–heart infusion broth (BHI, Becton Dickinson, Franklin Lakes, NJ, USA) supplemented with 10% bovine serum in GENbag CO_2_, BioMerieux (Lyon, France), at 37 °C for 48 h. *Staphylococcus aureus* ATCC6538, *Klebsiella pneumoniae* ATCC13883, *Escherichia coli* ATCC8739, *Salmonella enterica* ATCC13076, *Bacillus subtilis* ATCC6633, and *Bacillus cereus* ATCC11778 grew in Mueller–Hinton broth (cation-adjusted MH, Becton Dickinson, Franklin Lakes, NJ, USA) in an aerobic atmosphere at 37 °C for 48 h. *Helicobacter pylori* ATCC43504, *Campylobacter coli* ZMF, and *Campylobacter jejuni* ZMF grew in BHI supplemented with 10% bovine serum in microaerophilic atmosphere at 37 °C for 72 h (GENbag microaer, BioMerieux, Lyon, France). *Lactobacillus paracasei* PCM2639 and *Lactobacillus acidophilus* PCM2499 strains grew in De Man, Rogosa, and Sharpe (MRS) broth in GENbag CO_2_, BioMerieux (Lyon, France), at 37 °C for 48 h. After the determination of the bacterial viability, BHI blood, MH, and MRS agar plates were used.

Active cultures were prepared by transferring cells from the stock cultures to tubes with adequate broth as described above. They were incubated without agitation for 24 or 48 h at 37 °C. The cultures were diluted with adequate broth to achieve an optical density corresponding to 10^6^ colony forming units per mL (CFU/mL) for bacteria species (except *Helicobacter pylori*, *Campylobacter coli* ZMF, and *Campylobacter jejuni* ZMF). For *Helicobacter pylori*, *Campylobacter coli* ZMF, and *Campylobacter jejuni* ZMF, inoculum was prepared from bacterial colonies growing on BHI blood agar plates that had been incubated for 48 h in appropriate conditions. The final inoculum concentration was approximately 10^6^ CFU/mL [49,50,51]. The minimum inhibitory concentration (MIC) was determined by broth microdilution technique using 96-well plates. After filling each well with 100 μL of broth, dry test samples were dissolved in ethanol to a final concentration of 7 mg/mL. These solutions were diluted and added to the first well of each microtiter line. Dilutions in geometric progression were performed by transferring the mixture/dilution (100 μL) from the first to twelfth well. An aliquot (100 μL) was discarded from the twelfth well. The final concentration of yamogenin and reference ampicillin used in the antimicrobial assay ranged from 3.5 to 0.0006875 mg/mL and from 128 to 0.0625 μg/mL, respectively. The samples were incubated in adequate conditions at 37 °C for 48 h. The end point was determined by the visual observation of growth. The MIC values were considered as the lowest sample concentration that prevented visible growth. In addition, 100 μL of suspension from each well without growth was inoculated in agar plate to control bacterial viability. After 48 h of incubation, plates were checked for bacterial growth. The MBC (minimal bactericidal concentration) values were defined as the minimal concentration of the compounds required to kill of the organisms [51].

### 4.12. Statistical Analysis

Statistical data were obtained using the STATISTICA 12.0 software package (StatSoft Inc., Tulsa, OK, USA). All data were expressed as mean values ± standard deviation (±SD). The Student’s *t*-test was used to compare the results with the control sample. One-way ANOVA with Tuckey’s post hoc test was used to indicate significant differences among groups. The statistical significance was set at *p* < 0.05.

## 5. Conclusions

This study showed that yamogenin had cytotoxic activity on colon cancer HCT116 and gastric cancer AGS cells and this effect was dose-dependent. The combinations of yamogenin with oxaliplatin or capecitabine caused significantly stronger anticancer effect on the AGS cells compared to the activity of individual agents. Yamogenin induced cell death in gastric cancer cells via both the extrinsic and intrinsic pathway of apoptosis. Also, yamogenin as a plant compound may potentially be used in the treatment of gastric cancers and prevention of tumors development due to its moderate antioxidant and the inhibition of protein denaturation properties. However, these insights, shown for the first time, should be evaluated and confirmed in further in vivo studies.

## Figures and Tables

**Figure 1 ijms-25-04627-f001:**
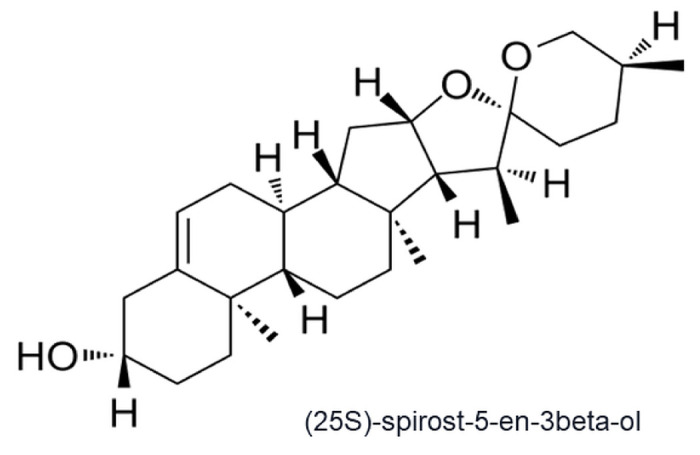
The chemical structure of yamogenin (neodiosgenin).

**Figure 2 ijms-25-04627-f002:**
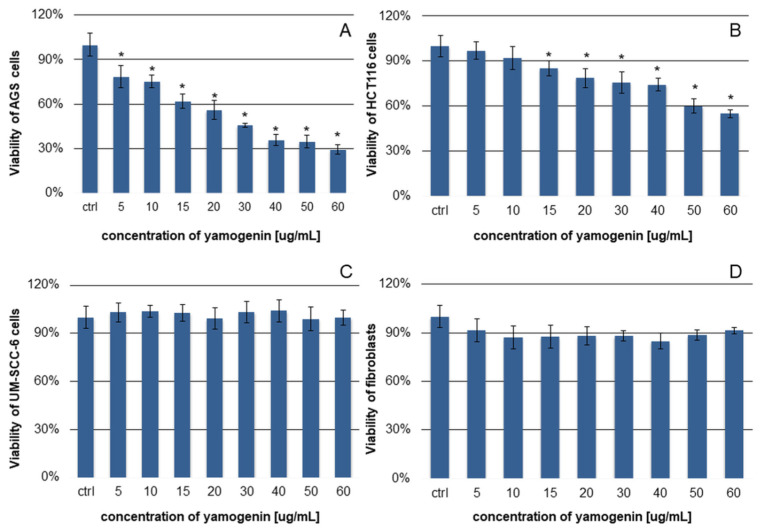
The cytotoxic effect of yamogenin on AGS (**A**), HCT116 (**B**), UM-SCC-6 cells (**C**), and fibroblasts (**D**). The cell lines were treated with yamogenin for 24 h at the concentrations of 5–60 µg/mL. The concentration of ethanol (ctrl) did not exceed 0.75% (*v*/*v*). The viability of the cells was estimated with MTT assay. The values represent the means ± standard deviations (±SD) obtained from two independent experiments in six repeats (*n* = 12). Error bars represent standard deviations. Significant differences relative to the control are marked with an asterisk (the Student’s *t*-test, * *p* < 0.05).

**Figure 3 ijms-25-04627-f003:**
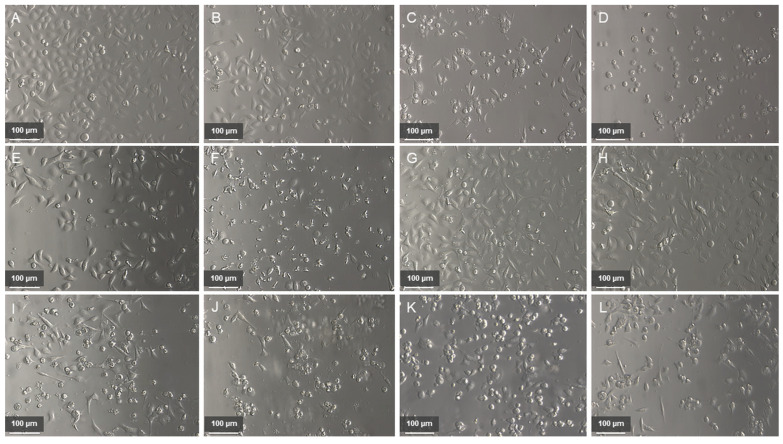
The AGS cells treated with yamogenin and in combination with oxaliplatin or capecitabine. The cells were incubated with ethanol (0.75%, *v*/*v,* ctrl—(**A**)), yamogenin at 10 (**B**), 30 (**C**), and 60 µg/mL (**D**), oxaliplatin at concentrations of 8 µg/mL (**E**) and 40 µg/mL (**F**), capecitabine at concentrations of 7.2 µg/mL (**G**) and 36 µg/mL (**H**), and in combinations: oxaliplatin and yamogenin (8 + 30 µg/mL, (**I**)), oxaliplatin and yamogenin (40 + 60 µg/mL, (**J**)), capecitabine and yamogenin (7.2 + 30 µg/mL, (**K**)), and capecitabine and yamogenin (36 + 60 µg/mL, (**L**)). The cells were observed under magnification 200× (Leica, Wetzlar, Germany).

**Figure 4 ijms-25-04627-f004:**
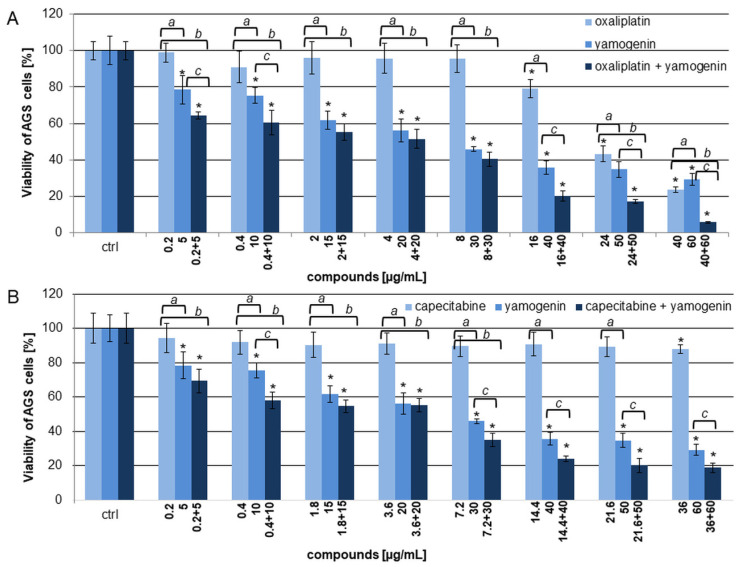
The cytotoxic effect of yamogenin in combination with oxaliplatin (**A**) or capecitabine (**B**). The AGS cells were treated with the compounds for 24 h at the concentrations of 5–60 µg/mL for yamogenin and 0.2–40 µg/mL for cytostatics, respectively. The concentration of ethanol (ctrl) did not exceed 0.75% (*v*/*v*). The viability of the AGS cells was obtained with MTT assay. The values represent the means ± standard deviations (±SD) obtained from three independent experiments in six repeats (*n* = 18). Error bars represent standard deviations. Significant differences relative to the control are marked with an asterisk (the Student’s *t*-test, * *p* < 0.05). Significant differences among groups are marked with *a*, *b*, and *c* (one-way ANOVA with Tuckey’s post hoc test, *p* < 0.05).

**Figure 5 ijms-25-04627-f005:**
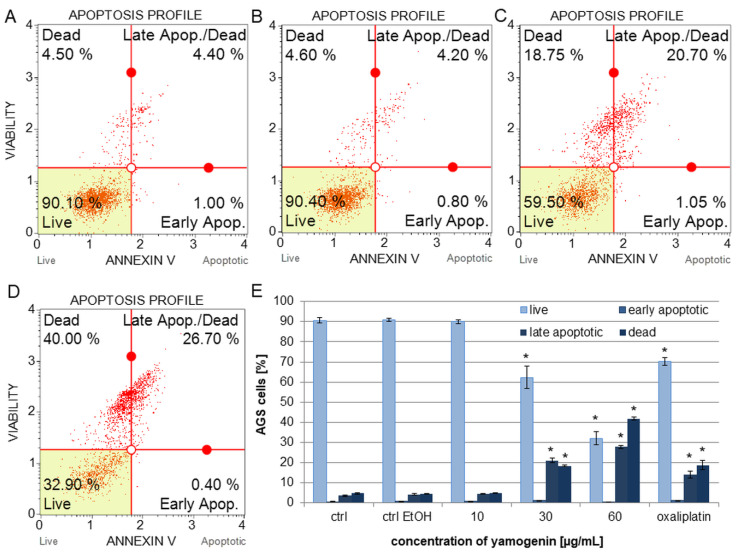
Apoptotic effect of yamogenin in the AGS cells. The cells were treated with ethanol (0.75%, *v*/*v*, ctrl EtOH, (**A**)) and the compound at the concentrations of 10 (**B**), 30 (**C**), and 60 µg/mL (**D**) for 24 h. Oxaliplatin was used as a positive control at a concentration of 40 µg/mL. The amount of live, apoptotic, and dead cells was estimated with flow cytometry. The values represent the means ± standard deviations (±SD) obtained from three independent experiments. Error bars represent standard deviations. Significant differences relative to the EtOH control are marked with an asterisk (the Student’s *t*-test, * *p* < 0.05) (**E**).

**Figure 6 ijms-25-04627-f006:**
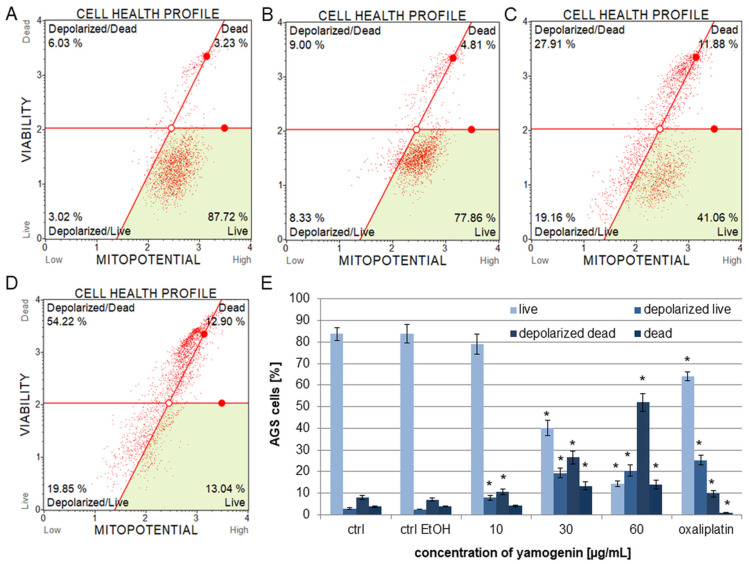
The effect of yamogenin on the MMP in the AGS cells. The cells were treated with ethanol (0.75%, *v*/*v*, ctrl EtOH, (**A**)) and yamogenin at the concentrations of 10 (**B**), 30 (**C**), and 60 µg/mL (**D**) for 24 h. Oxaliplatin was used as a positive control at a concentration of 40 µg/mL. The changes in the MMP were estimated with flow cytometry. The values represent the means ± standard deviations (±SD) obtained from three independent experiments. Error bars represent standard deviations. Significant differences relative to the EtOH control are marked with an asterisk (the Student’s *t*-test, * *p* < 0.05) (**E**).

**Figure 7 ijms-25-04627-f007:**
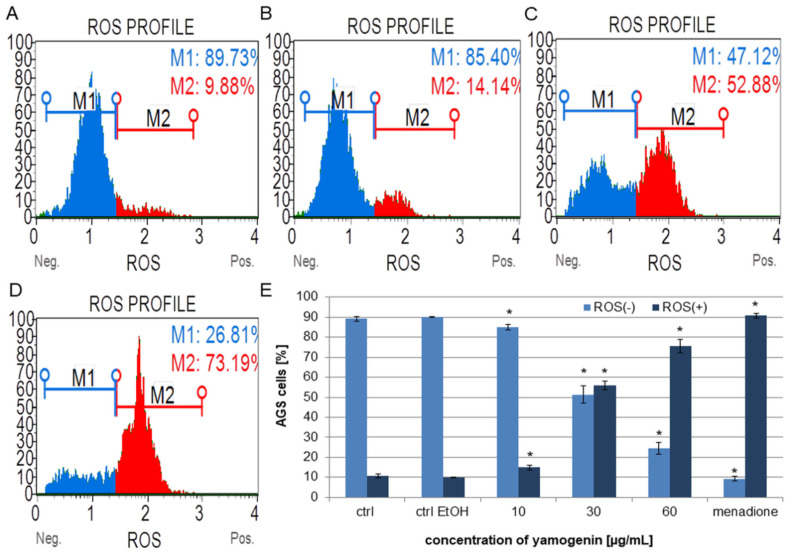
The effect of yamogenin on ROS production in the AGS cells. The cells were treated with ethanol (0.75%, *v*/*v*, ctrl EtOH, (**A**)) and yamogenin at the concentrations of 10 (**B**), 30 (**C**), and 60 µg/mL (**D**) for 24 h. Menadione was used as a positive control at a concentration of 17 µg/mL. The changes in the generation of oxidative stress were estimated with flow cytometry. The values represent the means ± standard deviations (±SD) obtained from three independent experiments. Error bars represent standard deviations. Significant differences relative to the EtOH control are marked with an asterisk (the Student’s *t*-test, * *p* < 0.05) (**E**).

**Figure 8 ijms-25-04627-f008:**
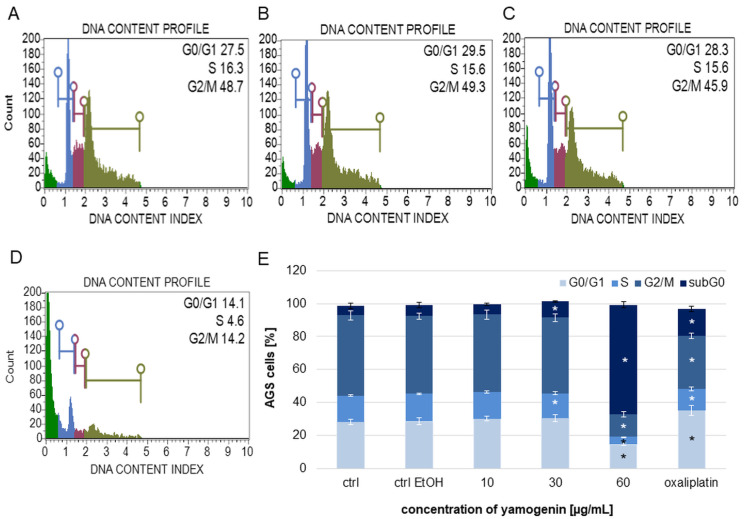
The effect of yamogenin on the inhibition of cell cycle in the AGS cells. The cells were treated with ethanol (0.75%, *v*/*v*, ctrl EtOH, (**A**)) and yamogenin at the concentrations of 10 (**B**), 30 (**C**), and 60 µg/mL (**D**) for 48 h and analyzed with flow cytometry. Green, blue, red, and khaki colors in (**A**–**D**) mean the cells in subG1, G0/G1, S, and G2/M phase, respectively. Oxaliplatin was used as a positive control at a concentration of 40 µg/mL. The values represent the means ± standard deviations (±SD) obtained from three independent experiments. Error bars represent standard deviations. Significant differences relative to the EtOH control are marked with an asterisk (the Student’s *t*-test, * *p* < 0.05) (**E**).

**Figure 9 ijms-25-04627-f009:**
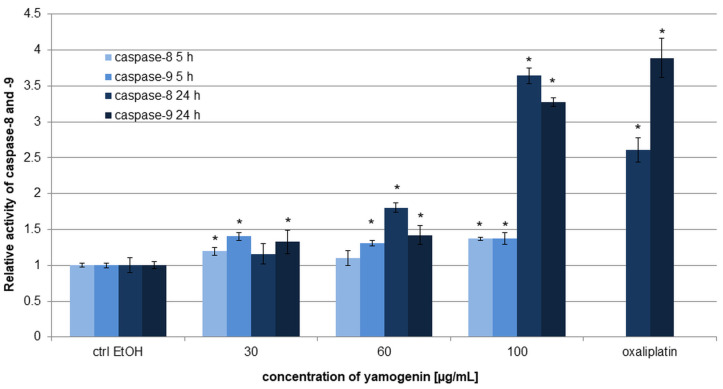
The effect of yamogenin on the activation of caspase-8 and -9 in the AGS cells. The cells were treated with yamogenin at the concentrations of 30–100 µg/mL for 5 and 24 h and analyzed with luminometry. The concentration of ethanol did not exceed 0.75% (*v*/*v*). Oxaliplatin was used as a positive control at a concentration of 20 µg/mL. The values represent the means ± standard deviations (±SD) obtained from three independent experiments. Error bars represent standard deviations. Significant differences relative to the EtOH control are marked with an asterisk (the Student’s *t*-test, * *p* < 0.05).

**Figure 10 ijms-25-04627-f010:**
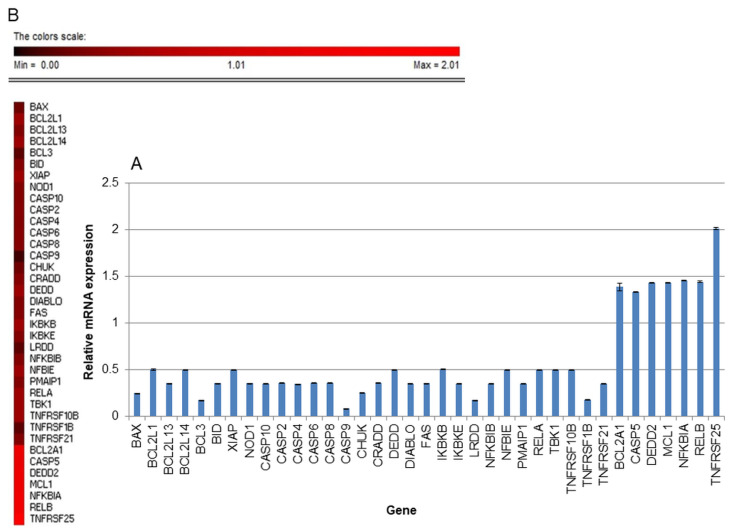
The relative gene expression at mRNA level in the AGS cells treated with yamogenin. The cells were incubated with ethanol (0.4% (*v*/*v*)—a control) and the sapogenin at a concentration of 30 μg/mL for 24 h. The results were obtained with Real-Time PCR, and the values represent the means ± standard deviations (±SD) of three independent experiments (**A**). Error bars represent standard deviations. The expression of the genes was normalized to 18S endogenous control gene, and their levels are shown as a fold change over the value 1.0 or under 0.5 (control). (**B**) Heatmap represents relative gene expression performed with PermutMatrix software v. 1.9.3 [18], where the results are expressed as a fold change over or under the value 1.0 (control) in the range of 0–2.

**Table 1 ijms-25-04627-t001:** Antiradical potential of yamogenin expressed as IC_50_ values (µg/mL) in DPPH and ABTS tests.

	IC_50_ µg/mL	
	Yamogenin	Ascorbic Acid
DPPH	704.7 ± 5.9	14.15 ± 0.13
ABTS	631.09 ± 3.51	7.33 ± 0.17

The results are presented as mean values ± standard deviations (±SD) and were obtained from three independent experiments with three repetitions (*n* = 9). The IC_50_ values of yamogenin differ significantly compared to the IC_50_ values of ascorbic acid (*p* < 0.05).

**Table 2 ijms-25-04627-t002:** The antimicrobial activity of yamogenin on bacteria strains.

Microorganism	[mg/mL]		Ampicillin [mg/mL]
	MIC	MBC	MIC
*Staphylococcus aureus* ATCC6538	0.35	>3.5	0.00008
*Klebsiella pneumoniae* ATCC13883	3.5	>3.5	0.001
*Escherichia coli* ATCC8739	3.5	>3.5	0.0039
*Salmonella enterica* ATCC13076	3.5	>3.5	0.0005
*Helicobacter pylori* ATCC43504	3.5	>3.5	0.0032
*Campylobacter jejuni* ZMF	3.5	>3.5	0.032
*Campylobacter coli* ZMF	3.5	>3.5	0.016
*Bacillus cereus* PCM 1948, 2019 (ATCC11778)	3.5	>3.5	0.125
*Listeria monocytogenes* PCM2191	3.5	>3.5	0.016
*Lactobacillus paracasei* PCM2639	>3.5	>3.5	>0.125
*Lactobacillus acidophilus* PCM2499	>3.5	>3.5	>0.125

MIC—minimum inhibitory concentration, MBC—minimal bactericidal concentration.

## Data Availability

Data are contained within the article.

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
