# Peer review of "An In Vitro Study on the Cytotoxic, Antioxidant, and Antimicrobial Properties of Yamogenin—A Plant Steroidal Saponin and Evaluation of Its Mechanism of Action in Gastric Cancer Cells"

_ijms, 2024, doi:10.3390/ijms25094627_

Round 1

Reviewer 1 Report

Comments and Suggestions for Authors

The manuscript submitted by Stefanowicz-Hajduk et al, entitled "Evaluation of the Cytotoxic Activity of Yamogenin on Human Cancer Cells" presents a study investigating the cytotoxic effects of yamogenin on human cancer cells, specifically gastric adenocarcinoma AGS, colorectal carcinoma HCT116, and squamous carcinoma UM-SCC-6 cell lines. They also explores the combination of yamogenin with cytostatics oxaliplatin and capecitabine in the treatment of gastrointestinal tract cancers. Overall, the manuscript is well-structured and provides valuable insights into the potential anticancer properties of yamogenin.

In general, the study could be of interest to our readers. I have a few comments on the study.

1. there seems to be a discrepancy between the experimental drug mentioned in the text, which is Yamogenin, and the provided keyword, which is neodiosgenin. Please verify the provided keywords.

2. The findings of the study align well with the conclusions that yamogenin may hold potential applications in the treatment of gastric cancers and in tumor prevention. However, further in vivo studies are recommended for validation.

3. Figures 4-7 lack explanations for the significance of each subplot. For example, the legend fails to clarify what 'A' represents. Providing clear legends for each subplot would enhance the interpretability of the figures.

4. there are some grammatical issues in the text that require revision, such as those evident in line 27-29. Please revise accordingly.

5. There are some errors in the reference citations, such as those found on line 297 and 305. These references need to be carefully revised for accuracy. Please ensure that all references are properly formatted and cited according to the author's guide.

Author Response

We would like to thank for critical reading this manuscript and valuable suggestions. We have carefully considered all of the suggestions and made the appropriate additions. The changes are marked in red in the text.

Reviewer I
Comment 1: There seems to be a discrepancy between the experimental drug mentioned in the text, which is Yamogenin, and the provided keyword, which is neodiosgenin. Please verify the provided keywords.

Response: Neodiosgenin is a synonym of yamogenin, this term has been added to our revised version of manuscript (please look at page 2, line 79 and Figure 1 on page 3, line 97).

Comment 2: The findings of the study align well with the conclusions that yamogenin may hold potential applications in the treatment of gastric cancers and in tumor prevention. However, further in vivo studies are recommended for validation.

Response: We agree with this comment that further in vivo studies regarding the effect of yamogenin on cancer are needed, especially in the context of anti-inflammatory and antioxidative potential of this natural steroid. These topics will be developed in a future project.

Comment 3: Figures 4-7 lack explanations for the significance of each subplot. For example, the legend fails to clarify what 'A' represents. Providing clear legends for each subplot would enhance the interpretability of the figures.

Response: Thank you for this comment. We have added the explanations for these figures (now are Figures 5, 6, 7, 8).

Comment 4: there are some grammatical issues in the text that require revision, such as those evident in line 27-29. Please revise accordingly.

Response: We have corrected the manuscript as much as possible, also this fragment from the abstract has been changed.

Comment 5: There are some errors in the reference citations, such as those found on line 297 and 305. These references need to be carefully revised for accuracy. Please ensure that all references are properly formatted and cited according to the author's guide.

Response: Thank you. We have corrected this.

 Once again, thank you for your time and valuable comments.

Sincerely,

Justyna Stefanowicz-Hajduk

Reviewer 2 Report

Comments and Suggestions for Authors

General Comments:

The manuscript by Justyna, et al experimentally demonstrated the wide spectrum of biological activities of yamogenin on human cancer cells, particularly focusing on gastrointestinal tract cancers. The current manuscript provides experimental evidence into potential therapeutic agents derived from natural sources. Below are some critical comments to be considered for improvement or further investigation.

Major Comments:

1st, The rationale of in vitro cell model selections: the authors had particular interests in gastrointestinal tract cancers. They chose human colon cancer HCT116, gastric cancer AGS, squamous carcinoma UM-SCC-6 cells, and hu-18 man normal fibroblasts for a cytotoxicity screening and performed the rest of the assay on a single cell line (gastric cancer AGS). In considering the representativeness of cancer cell lines to clinical feature and mutation landscape gastric cancer, at least, 2 to 3 cell lines that capture the most actionable or targetable mutations (TP53, KRAS, PIK3, CTNNB1, et al) should be used for cell-lines based studies. Please refer to the ATCC Stomach (Gastric) Cancer Panel (ATCC® TCP-1008™) for reference.

2nd, In the manuscript, the yamogenin's effects are clearly outlined, however, a direct comparison with other known and defined anticancer compounds that could generate clear effects should be included as reference control, especially other steroid saponins, would provide a clearer context for its efficacy. This could help position yamogenin among potential natural anticancer agents. For instance, in Fig.6, the authors used “menadione, " known to generate (ROS) through redox cycling as a Reference control. It is recommended to apply such a design to other assays.

3rd, In Fig 9, the study indicates significant changes in the expression of apoptosis-related genes but does not dig into the implications of these changes. Future studies should consider a more comprehensive analysis of gene expression changes, potentially through RNA-seq, to understand the global impact of yamogenin on cancer cell gene expression networks.

4th, In section 2.8, the study briefly touches on the antioxidant potential of yamogenin by performing a biochemistry assay, which was lack of depth at the cell level. Given the importance of inflammation and oxidative stress in cancer progression, more detailed studies with cells should be designed to reveal such benefits of yamogenin.

Minor Comments:

1st , Fig.2, the background of widefield microscopic images are high, adjustment on brightness would be necessary. Alternatively, it also recommended to apply fluorescent labels to identify dead cells, to identify live cells.

2nd, Fig 9. It is suggested to use different plots to demonstrate differential gene expression, for example, a heatmap including individual treatments should be considered.

3rd, In discussion, the manuscript would benefit from a discussion about the limitations and shortcomings of the current manuscript, particularly the lack of in vivo studies. In vivo studies are essential to evaluate the compound's efficacy and safety in a more complex biological context.

Author Response

Reviewer II

Dear Editor and Reviewer,
We would like to thank for critical reading this manuscript and valuable suggestions. We have carefully considered all of the suggestions and made the appropriate additions. The changes are marked in red in the text.

Comment 1: The rationale of in vitro cell model selections: the authors had particular interests in gastrointestinal tract cancers. They chose human colon cancer HCT116, gastric cancer AGS, squamous carcinoma UM-SCC-6 cells, and hu-18 man normal fibroblasts for a cytotoxicity screening and performed the rest of the assay on a single cell line (gastric cancer AGS). In considering the representativeness of cancer cell lines to clinical feature and mutation landscape gastric cancer, at least, 2 to 3 cell lines that capture the most actionable or targetable mutations (TP53, KRAS, PIK3, CTNNB1, et al) should be used for cell-lines based studies. Please refer to the ATCC Stomach (Gastric) Cancer Panel (ATCC® TCP-1008™) for reference.

Response: Thank you for this comment. We agree with this and we plan to buy and use such a panel of cell lines in our future studies concerning the activity of the natural compounds on gastric cancer.  

Comment 2: In the manuscript, the yamogenin's effects are clearly outlined, however, a direct comparison with other known and defined anticancer compounds that could generate clear effects should be included as reference control, especially other steroid saponins, would provide a clearer context for its efficacy. This could help position yamogenin among potential natural anticancer agents. For instance, in Fig.6, the authors used “menadione, " known to generate (ROS) through redox cycling as a Reference control. It is recommended to apply such a design to other assays.

Response: We have made additionally experiments and added as a reference control oxaliplatin, which was in fact used in our experiments in this project. Please look at Figures 5, 6, 8, and 9.

Comment 3: In Fig 9, the study indicates significant changes in the expression of apoptosis-related genes but does not dig into the implications of these changes. Future studies should consider a more comprehensive analysis of gene expression changes, potentially through RNA-seq, to understand the global impact of yamogenin on cancer cell gene expression networks.

Response: We agree with this comment and we will take this into account in our future research.

Comment 4: In section 2.8, the study briefly touches on the antioxidant potential of yamogenin by performing a biochemistry assay, which was lack of depth at the cell level. Given the importance of inflammation and oxidative stress in cancer progression, more detailed studies with cells should be designed to reveal such benefits of yamogenin.

Response: In our work we show only preliminary results regarding the biological activity of yamogenin and we want to highlight the need to continue these studies, especially in the context of anti-inflammatory and antioxidative potential of this natural steroid, also in in vivo research. These topics will be developed in a future project.

Comment 5: Fig.2, the background of widefield microscopic images are high, adjustment on brightness would be necessary. Alternatively, it also recommended to apply fluorescent labels to identify dead cells, to identify live cells.

Response: We have improved our images, adjusted the brightness and sharpened every image (now is Figure 3, page 4). Preparing the experiment with fluorescent dyes was impossible for us in such a short time because we would have to wait too long for the delivery of reagents.

Comment 6: Fig 9. It is suggested to use different plots to demonstrate differential gene expression, for example, a heatmap including individual treatments should be considered.

Response: According to this comment, we have done and added a heatmap prepared with PermutMatrix software (now this is Figure 10, page 11).

Comment 7: In discussion, the manuscript would benefit from a discussion about the limitations and shortcomings of the current manuscript, particularly the lack of in vivo studies. In vivo studies are essential to evaluate the compound's efficacy and safety in a more complex biological context.

Response: We have added this fragment to our manuscript (please look at page 14, lines 394-401).

Once again, thank you for your time and valuable comments.

Sincerely,

Justyna Stefanowicz-Hajduk

Reviewer 3 Report

Comments and Suggestions for Authors

The authors have performed a systematic in vitro study investigating the effects of yamogenin on gastric cancer cells and gave insight into its mechanism of inducing apoptosis. The manuscript is well written, and the authors have employed a combination of characterization methods to support their findings. I only have some minor issues to further improve this manuscript.

1. The chemical structure of yamogenin should be included in the main text.

2. Lines 72-73, any evidence to support “natural drugs are more safe than synthetic drugs”? This statement is misleading.

3. Figure 4, To improve clarity, each subfigure should be briefly described in the caption.

4. Figure 8, the compound concentration shown from 30 to 100 in the figure which is not consistent with the caption described.

5. lines 297 and 306, citation format appears to be incorrect.

Author Response

Reviewer III

Dear Editor and Reviewer,
We would like to thank for critical reading this manuscript and valuable suggestions. We have carefully considered all of the suggestions and made the appropriate additions. The changes are marked in red in the text.

Comment 1: The chemical structure of yamogenin should be included in the main text.

Response: The structure of yamogenin has been added to the manuscript (Figure 1, page 3).

Comment 2: Lines 72-73, any evidence to support “natural drugs are more safe than synthetic drugs”? This statement is misleading.

Response: Thank you for this comment. We agree and have deleted this fragment (page 2, line 74).

Comment 3: Figure 4, To improve clarity, each subfigure should be briefly described in the caption.

Response: We have added this description to the caption of Figure 5 (before it was Fig. 4, page 6).

Comment 4: Figure 8, the compound concentration shown from 30 to 100 in the figure which is not consistent with the caption described.

Response: Thank you. We have corrected this description (now Figure 9, page 10).

Comment 5: lines 297 and 306, citation format appears to be incorrect.

Response: We have corrected this.

Once again, thank you for your time and valuable comments.

Sincerely,

Justyna Stefanowicz-Hajduk

Round 2

Reviewer 2 Report

Comments and Suggestions for Authors

The authors have enhanced the current version substantially by incorporating additional data and expanding the discussion.

They have acknowledged the challenges in performing several key experiments and have discussed potential future directions, which is acceptable.

However, from a scientific standpoint, it is essential to conduct in vitro studies using a variety of cell types to reflect differences in pathology and responses to drugs.

Author Response

We would like to thank for critical reading this manuscript and valuable suggestions.

Comments: The authors have enhanced the current version substantially by incorporating additional data and expanding the discussion.

They have acknowledged the challenges in performing several key experiments and have discussed potential future directions, which is acceptable.

However, from a scientific standpoint, it is essential to conduct in vitro studies using a variety of cell types to reflect differences in pathology and responses to drugs.

Response: We are grateful for your valuable comments which will be taken into account in subsequent studies on yamogenin and gastric cancer. The research described in this manuscript concerns the determination of the cytotoxic effect of the sapogenin on selected cell lines of gastrointestinal tract and indicates the direction of further research in future projects, in which other gastric cancer cell lines (also isolated from metastatic sites) will be used.

Once again, thank you for your time and valuable comments.

Sincerely,
Justyna Stefanowicz-Hajduk
Assistant Professor
Medical University of Gdansk